# A Preclinical Feasibility Study of Single-Port Robotic Subcostal Anatomical Lung Resection and Subxiphoid Thymectomy Using the da Vinci^®^ SP System

**DOI:** 10.3390/diagnostics13030460

**Published:** 2023-01-26

**Authors:** Ching Feng Wu, Chuan Cheng, Ka Hei Suen, Hubert Stein, Yin Kai Chao

**Affiliations:** 1Division of Thoracic Surgery, Chang Gung Memorial Hospital at Linkou, Chang Gung University College of Medicine, Taoyuan 333323, Taiwan; 2Division of Thoracic Surgery, New Taipei Municipal Tu-Cheng Hospital, New Taipei City 25162, Taiwan; 3Department of Surgical Applications Engineering, Intuitive Surgical Inc., Sunnyvale, CA 94086, USA

**Keywords:** da Vinci^®^ SP system, single-port thoracic surgery, subxiphoid surgery, subcostal surgery

## Abstract

Despite the recent introduction of technologically advanced single-port (SP) robotic systems, their use in the field of thoracic surgery has been rarely explored. Here, we report our preclinical experience concerning SP robotic thoracic surgery using the da Vinci^®^ SP system. The da Vinci^®^ SP system was used to perform subcostal anatomical lung resection and subxiphoid thymectomy in three cadavers. The operative settings that best met the surgeon’s requirements for each resection were also determined. Four subcostal anatomical lung resections and two subxiphoid thymectomies were completed. While both procedures did not require additional incisions, the use of an observation port in the intercostal spaces was strongly recommended to safely create subcostal access. Dissection of hilar structures and mediastinal lymph nodes was feasible. However, due to the current unavailability of a robotic stapler, a handheld stapling instrument was required to perform a transection of vital structures. When the stapling process proved to be difficult, the table surgeon temporarily removed a robotic arm to acquire the necessary space to complete the procedure. Our data represent a promising preclinical step in understanding the feasibility of using the da Vinci^®^ SP system to perform an SP subcostal anatomical lung resection and a subxiphoid thymectomy.

## 1. Introduction

After three decades from the first description of thoracoscopic lobectomy by Roviaro et al. in 1992, video-assisted thoracoscopic surgery (VATS) has become a common and globally accepted surgical procedure [1,2]. In recent years, two major trends have been dominant in the area of minimally invasive thoracic surgery. First, several reports have shown that single-port (SP) VATS is feasible and provides comparable short-term outcomes to the traditional procedure, while allowing for a reduced number of incisions [3,4]. However, suboptimal ergonomics continues to challenge the widespread adoption of this approach, especially in the case of complex procedures. Second, robot-assisted thoracoscopic surgery has gained increasing traction due to major outcome advantages—including better ergonomics and an increased dissection precision, while maintaining minimal invasiveness [5,6]. On a global scale, the da Vinci^®^ surgical system (Intuitive Surgical; Sunnyvale, CA, USA) remains the most commonly used platform for robotic surgery. However, the conventional approach still requires the application of multiple (4–5) skin incisions for the robotic arms. The recent advent of SP robotic surgery came with the promise that the two trends were converging, and the recent da Vinci^®^ SP platform (Intuitive Surgical) has been developed to integrate the advantages of single-port and robotic surgeries. The robot provides three multi-jointed wristed instruments as well as a fully wristed high-definition (HD) three-dimensional (3D) stereoscopic binocular camera through a single 2.8 cm cannula. Although the United States Food and Drug Administration (FDA) has currently cleared the da Vinci^®^ SP platform for urology and transoral otolaryngology procedures, multiple clinical studies in the United States and Taiwan are currently investigating its potential usefulness in the field of thoracic surgery [7,8,9]. In this preclinical cadaveric study, we report our preliminary feasibility experience in the field of SP robotic thoracic surgery using the da Vinci^®^ SP system. We specifically examined whether this technique may represent a valid approach to subcostal anatomical lung resection and subxiphoid thymectomy.

## 2. Materials and Methods

### 2.1. Overview of the da Vinci^®^ SP System

The da Vinci^®^ SP system has three primary components: (1) the patient-side cart, which consists of a single arm that controls up to three multi-jointed wristed instruments and the fully wristed HD 3D articulating camera; (2) the vision system; and (3) a free-standing surgeon’s console. Similar to other da Vinci^®^ platforms (Xi and X), the wrist-like robotic arm is designed to provide seven degrees of freedom of motion during the procedure (i.e., rotation, in–out, pitch, yaw, grasp, wristed pitch, and wristed yaw).

### 2.2. Study Setting

All the procedures were carried out in a third-party laboratory located in Germany. The da Vinci^®^ SP system was used to perform four subcostal anatomical lung resections (three lobectomies and one segmentectomy) and two subxiphoid thymectomies in three fresh cadavers. Pre-surgical training of the two surgeons performing the procedures consisted of formal lectures and didactic sessions. During surgery, operative details were thoroughly discussed and captured with image and video recordings. The study protocol followed the ethical principles outlined in the Declaration of Helsinki. Preclinical studies on cadavers do not fall under the technical definition of human subject research; therefore, the study did not require Institutional Review Board review and approval.

### 2.3. Uniportal Subxiphoid Thymectomy

After placing the cadaver in the supine position, a 3.5 cm transverse incision was made approximately 3 cm caudal to the xiphoid process (Figure 1a). After incising the linea alba, a blunt finger dissection was performed to obtain sufficient space in the preperitoneal and retrosternal areas. The lower portions of the mediastinal pleura were subsequently subjected to bilateral detachment. After applying the uniportal access platform (da Vinci^®^ SP Access Port Kit; Figure 1b), the single 2.5 cm trocar was inserted, connected to an insufflator, and docked to the da Vinci^®^ SP patient-side cart arm (Figure 1c). The camera was inserted through the endoscope specific lumen (upper middle), whereas a fenestrated bipolar forceps was placed into the left lumen (arm #1). A Cadiere forceps and a monopolar scissor were inserted through the lower-middle lumen (arm #2) and the right lumen (arm #3), respectively.

### 2.4. Uniportal Subcostal Anatomical Lung Resection

The subcostal approach was performed with the cadaver placed in the lateral decubitus position. A 10 mm observation port was initially created in the sixth intercostal space (ICS) along the posterior axillary line. The thoracic cavity was subsequently insufflated with CO_2_ to a pressure of 8 mm Hg to promote inferior diaphragmatic displacement (Figure 2a). After performing a 4 cm skin incision immediately below the subcostal margin at the mid-clavicular line, the subcutaneous tissue and the oblique muscles were incised until the transverse abdominis fascia was visible. The pleura was accessed by tunneling below the costal cartilages and above the diaphragm using fingers and long Metzenbaum scissors under thoracoscopic guidance (Figure 2b). During surgery, the diaphragm was protected by preemptively suturing the cut edge of the diaphragmatic parietal pleura to the transverse abdominis fascia—followed by the application of the uniportal access device (da Vinci^®^ SP Access Port Kit). The single 2.5 cm trocar was subsequently inserted, connected to an insufflator, and docked to the da Vinci^®^ SP patient-side cart arm. The camera was inserted through the upper-middle lumen, whereas a Maryland bipolar forceps was placed into the left lumen (arm #1). A Cadiere forceps and a monopolar scissor were inserted through the lower-middle lumen (arm #2) and the right lumen (arm #3), respectively.

## 3. Results

### 3.1. Uniportal Subxiphoid Thymectomy

The sequence of dissection used for SP subxiphoid thymectomy was similar to that previously reported for the da Vinci^®^ Xi system—with the obvious exception of the absence of a bilateral intercostal incision [10]. While the use of a third robotic arm allowed a meticulous dissection along the innominate vein (Figure 3), dissection over the cardiophrenic angle proved to be more difficult compared with the multi-port Xi system. In particular, full articulation of the instrument was unfeasible due to the short distance between the target and the cannula tip. While the da Vinci^®^ SP Access Port Kit partly enhanced wrist articulation, the dissection required very fine and cautious movements to avoid collision.

### 3.2. Uniportal Subcostal Anatomical Lung Resection

The sequence of dissection used for robotic subcostal lobectomy/segmentectomy—which was similar to that applied for the conventional intercostal robotic technique—depended on the surgeon’s preferences and the degree of pulmonary fissure completeness. Camera positioning was guided by the target anatomy and the desired counter-traction force. Dissection was mainly performed using a monopolar scissor and a Maryland bipolar forceps. The target pulmonary artery and vein were encircled with a vessel loop (Figure 4a). In-depth surgical dissection along the vessels was required to obtain adequate space for safe introduction of the stapler (Figure 4b). The insertion of the handheld endovascular stapling instrument was achievable from the subcostal incision through the assistant port of the da Vinci^®^ SP Access Port Kit (Figure 4c). To achieve a sufficient space for stapling, removal of a robotic arm was sporadically required (arm #1 or arm #3 for left- and right-sided procedures, respectively). During the entire process, the console surgeon and the table surgeon were required to maintain effective communication to avoid collisions between the stapler and the robotic arm. Upon completion of the procedure, the proximal end of an oblique muscle, the cut edge of the diaphragmatic parietal pleura, and the distal end of an oblique muscle were sutured together; the rest of the wound was closed with one chest tube placed over the edge of the subcostal incision.

## 4. Discussion

The dawn of single-incision robotic surgery started with the introduction of the da Vinci^®^ Single-Site^®^ technology [11,12]. However, this platform did not feature articulating instruments and did not allow widespread adoption of the SP approach in thoracic surgery [11]. To our knowledge, published data are limited to thymectomy only [13,14]. The da Vinci^®^ SP system has great potential to be at the forefront of developing novel SP surgical applications. First, this model is equipped with flexible instruments that overcome restrictions of movement and may prove advantageous for meticulous dissections. Second, the parallel entrance of the instruments helps alleviate the collision problem. Third, this system includes three jointed instruments that allow completing highly challenging procedures. Starting from these premises, we designed the current preclinical study to test the feasibility of a robotic subcostal anatomical lung resection and a subxiphoid thymectomy performed by means of the da Vinci^®^ SP system.

Although the subxiphoid approach is generally considered an ideal minimally invasive approach for an anterior mediastinal lesion, problems related to ergonomic conditions—caused by interferences between the camera and non-articulating instruments—and difficulties in manipulation still represent a considerable challenge during the course of VATS [15,16]. The use of the da Vinci^®^ Si or Xi platforms can partially overcome these shortcomings, but does not obviate the need to resort to two intercostal incisions [10,17]. In our preclinical cadaveric study, the use of the da Vinci^®^ SP system allowed avoiding the intercostal port access, while ensuring a meticulous dissection through the uniportal access. These advantages have the potential for maximizing the safety and efficacy of a minimally invasive subxiphoid approach.

While the da Vinci^®^ SP system holds promise for becoming the standard of care for subxiphoid surgery, its application for transthoracic anatomical lung resection requires further discussion. Given that the minimum diameter needed for the cannula (2.8 cm) makes insertion in the ICS unfeasible, entry in the pleural cavity is currently achievable via the subcostal route only—an approach which is attainable, but less familiar to thoracic surgeons. In general, the use of a subcostal route is expected to offer significant advantages, including simple-specimen retrieval and the ability to spare the intercostal nerves. This would in turn reduce the risk of chronic postoperative pain—one of the most feared long-term complications of the intercostal approach. Unfortunately, there are still two major obstacles to the adoption of a subcostal access during the course of VATS. First, most VATS instruments are rigid and cannot be angled. In this scenario, the instrument’s range of motion might not be sufficient to cover the entire distance between the subcostal arch and the hilum. Second, the rigidity of the camera ultimately precludes a superior-to-inferior view. The introduction of the da Vinci^®^ SP system has helped to overcome these issues. Accordingly, the robot is equipped with three multi-jointed wristed instruments that allow a thorough dissection of both hilar structures and mediastinal lymph nodes, without resorting to the help of an assistant. In addition, the availability of a fully-wristed binocular camera with the concomitant use of the “cobra” mode markedly enhances surgical visualization.

Although the use of the da Vinci^®^ SP system has the potential to create a new standard of care for surgical treatment of early lung cancer, technical refinements to the transection of vital structures during anatomical lung resection are still required. In this regard, the lack of high-energy systems (i.e., robotic staplers or vessel sealers) compatible with the da Vinci^®^ SP system remains a major hurdle. The use of a handheld stapling instrument by a technically skilled bedside assistant was therefore a key prerequisite to perform bronchial or major vessel transection. It is also worth noting that staple insertion through a single incision was not invariably feasible and we were not always able to avoid collisions between the staple and the robotic arm. When stapling around the intended vascular site proved to be difficult, a multistep procedure was implemented. First, following an extensive dissection, pulmonary vessels were encircled with a vessel loop. Second, the robotic arm more closely located to the table surgeon was removed (i.e., arm #1 or arm #3 during left- and right-side procedures, respectively). Third, the staple (preferably with curved tip design) was advanced into the visual field together with the robotic instrument. Fine adjustment of the staple angle was subsequently performed under a robotic camera view. Once the adjustment was completed, the staple was advanced by the assistant as close as possible to the target site. Finally, traction on the lobe and the vessel loop was adjusted in the direction of the stapler tip to promote its smooth passing.

As for lymph node dissection, the flexibility of the SP robotic arm enabled a meticulous and complete dissection throughout every station—the only exception being the upper mediastinum in certain cases. Owing to the limited length of the instrument, the dissection of the most cranial station (Gr2 over the right side) was not invariably feasible; special challenges were observed when the thorax had a high vertical length (measured from the subcostal incision to the apical area). Potential future solutions to this issue include the development of longer instruments and/or the adoption of a smaller port device to facilitate entry through the intercostal space.

This study represents exploratory work and, as such, there are several limitations that need to be considered. While our findings open interesting opportunities for clinical human applications, the feasibility and safety of SP robotic thoracic surgery for subcostal anatomical lung resection and subxiphoid thymectomy remain to be confirmed in the living patient. In this regard, pulsating vessels and the risk of bleeding are expected to introduce additional procedural challenges. Additionally, cardiac motion could make this operating method more technically demanding—especially when a left subcostal approach is required. The use of the da Vinci^®^ SP platform for thoracic surgery is not yet approved by the FDA. Several clinical trials (NCT05455840/NCT05535712) are currently ongoing to evaluate the safety and clinical outcomes of this device for surgical treatment of different thoracic diseases using either the subxiphoid or the subcostal approach. Initial results are expected in August 2023.

## 5. Conclusions

Our data represent a promising preclinical step in understanding the feasibility of using the da Vinci^®^ SP system to perform single-port subcostal anatomical lung resection and subxiphoid thymectomy. While both procedures did not require additional incisions, the use of an observation port in the intercostal spaces was strongly recommended to guide the creation of subcostal access. Further research is needed to clarify the potential clinical usefulness of this approach in the field of thoracic surgery.

## Figures and Tables

**Figure 1 diagnostics-13-00460-f001:**
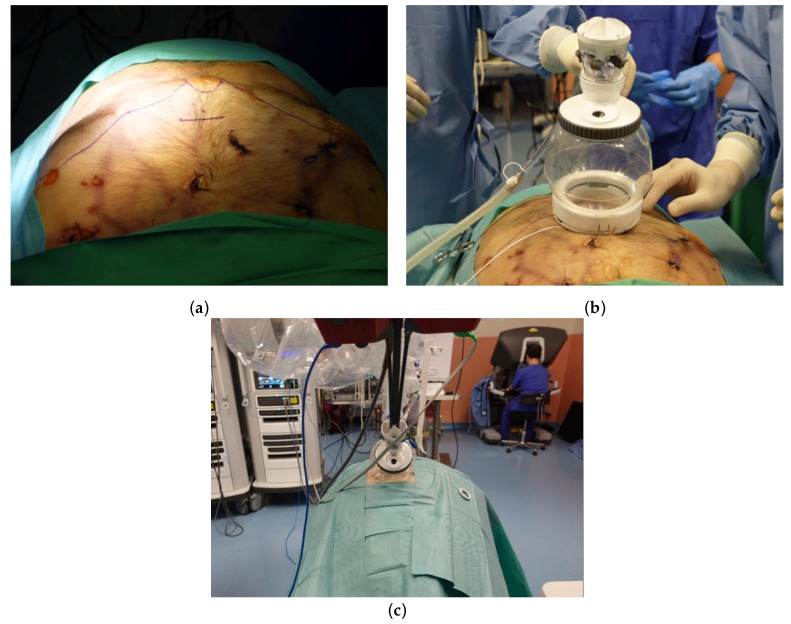
Subxiphoid thymectomy performed by means of the da Vinci^®^ SP system. (**a**) Incision design: a 3 cm incision was created approximately 2 cm caudal to the xiphoid process; (**b**) access port for the da Vinci^®^ SP platform; (**c**) whole view from the patient cart and the console.

**Figure 2 diagnostics-13-00460-f002:**
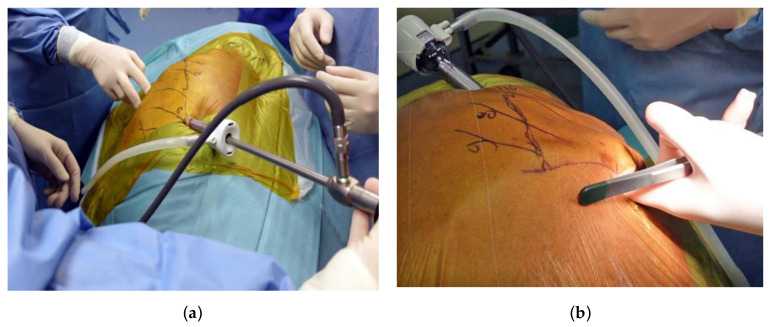
Subcostal anatomical lung resection performed by means of the da Vinci^®^ SP system. (**a**) An intercostal observation port was created in the sixth intercostal space along the middle-axillary line; (**b**) a 4 cm incision was initially made along the mid-clavicular line immediately below the subcostal margin, followed by the creation of a tunnel via finger blunt dissection under thoracoscopic guidance.

**Figure 3 diagnostics-13-00460-f003:**
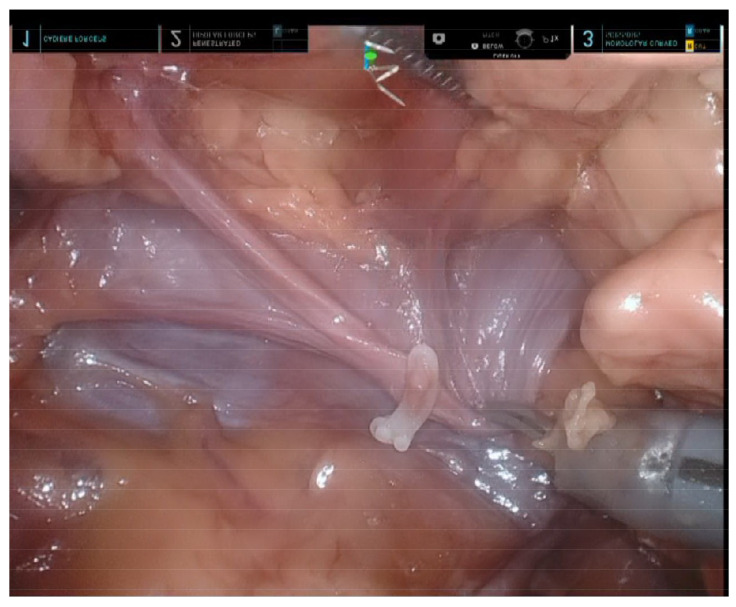
Final view after completion of subxiphoid thymectomy performed by means of the da Vinci^®^ SP system. The innominate vein underwent complete skeletonization, whereas the thymic vein branch was controlled using Hem-o-Lok clips.

**Figure 4 diagnostics-13-00460-f004:**
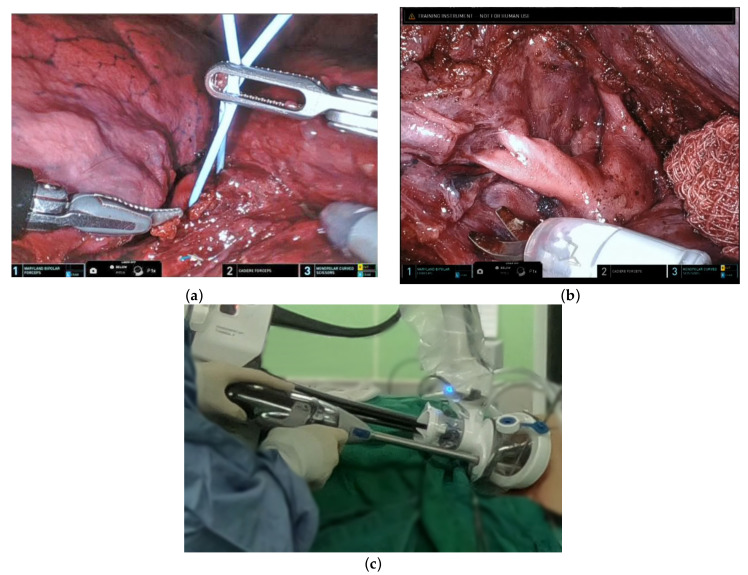
The target pulmonary artery was encircled with a vessel loop. (**a**) Expansion of the working space to allow safe entering of the staple was achieved through (**b**) an extensive dissection along the pulmonary vessels. (**c**) The stapler was introduced through the assistant port of the da Vinci^®^ SP Access Port Kit via a subcostal incision.

## Data Availability

Not applicable.

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
