# Peer review of "A Preclinical Feasibility Study of Single-Port Robotic Subcostal Anatomical Lung Resection and Subxiphoid Thymectomy Using the da Vinci® SP System"

_diagnostics, 2023, doi:10.3390/diagnostics13030460_

Round 1

Reviewer 1 Report

I understand that this is pioneering work in thoracic surgery area. Limited data in thymectomy and lobectomy preclude the evaluation of safety and effectiveness compared to existing multiport technology. I would also like to see some discussion as to the technique and evaluation of lymph node assessment wether that’s sampling or dissection. Also details regarding diaphragm closure at subcostal incision. Port that stapler comes in through as well as placement of chest tube site. Should all be included. 

Author Response

I understand that this is pioneering work in thoracic surgery area. Limited data in thymectomy and lobectomy preclude the evaluation of safety and effectiveness compared to existing multiport technology. I would also like to see some discussion as to the technique and evaluation of lymph node assessment whether that’s sampling or dissection.

<Response>

Thanks for your comment. Added per suggestion. As for lymph node dissection, the flexibility of the SP robotic arm enabled a meticulous and complete dissection throughout every station – the only exception being the upper mediastinum in certain cases. Owing to the limited length of the instrument, the dissection of the most cranial station (Gr2 over the right side) was not invariably feasible; special challenges were observed when the thorax had a high vertical length (measured from the subcostal incision to the apical area). Potential future solutions to this issue include the development of longer instruments and/or the adoption of a smaller port device to facilitate entry through the intercostal space.

Also details regarding diaphragm closure at subcostal incision. Port that stapler comes in through as well as placement of chest tube site. Should all be included. 

<Response>

Thanks for your comment. Added per suggestion.

The insertion of the handheld endovascular stapling instrument was achievable from the subcostal incision through the assistant port of the da Vinci® SP Access Port Kit (Figure 4C). To achieve a sufficient space for stapling, removal of a robotic arm was sporadically required (arm #1 or arm #3 for left and right-sided procedures, respectively). During the entire process, the console surgeon and the table surgeon were required to maintain effective communication to avoid collisions between the stapler and the robotic arm. Upon completion of the procedure, the proximal end of an oblique muscle, the cut edge of the diaphragmatic parietal pleura, and the distal end of an oblique muscle were sutured together; the rest of the wound was closed with one chest tube placed over the edge of the subcostal incision.

Reviewer 2 Report

The article is well written and gives an interesting insight in a field that has not been explored in a sufficient way until now. This could be a good first step for further researches.

Author Response

The article is well written and gives an interesting insight in a field that has not been explored in a sufficient way until now. This could be a good first step for further researches.

<Response>

Thanks for your positive comments!